# Inhibition of High-Temperature Requirement Protein A2 Protease Activity Represses Myogenic Differentiation via UPRmt

**DOI:** 10.3390/ijms231911761

**Published:** 2022-10-04

**Authors:** Hongyu Sun, Luyan Shen, Ping Zhang, Fu Lin, Jiaoyan Ma, Ying Wu, Huimei Yu, Liankun Sun

**Affiliations:** Department of Pathophysiology, College of Basic Medical Sciences, Jilin University, Changchun 130021, China

**Keywords:** HtrA2/Omi, mitonuclear imbalance, UPRmt, sarcopenia, skeletal muscle

## Abstract

Skeletal muscles require muscle satellite cell (MuSC) differentiation to facilitate the replenishment and repair of muscle fibers. A key step in this process is called myogenic differentiation. The differentiation ability of MuSCs decreases with age and can result in sarcopenia. Although mitochondria have been reported to be involved in myogenic differentiation by promoting a bioenergetic remodeling, little is known about the interplay of mitochondrial proteostasis and myogenic differentiation. High-temperature-requirement protein A2 (HtrA2/Omi) is a protease that regulates proteostasis in the mitochondrial intermembrane space (IMS). Mice deficient in HtrA2 protease activity show a distinct phenotype of sarcopenia. To investigate the role of IMS proteostasis during myogenic differentiation, we treated C2C12 myoblasts with UCF101, a specific inhibitor of HtrA2 during differentiation process. A key step in this process is called myogenic differentiation. The differentiation ability of MuSCs decreases with age and can result in sarcopenia. Further, CHOP, p-eIF2α, and other mitochondrial unfolded protein response (UPRmt)-related proteins are upregulated. Therefore, we suggest that imbalance of mitochondrial IMS proteostasis acts via a retrograde signaling pathway to inhibit myogenic differentiation via the UPRmt pathway. These novel mechanistic insights may have implications for the development of new strategies for the treatment of sarcopenia.

## 1. Introduction

As the body ages, the most common changes in skeletal muscles are loss of mass and loss of function. Muscle degeneration can develop into sarcopenia, leading to ad-verse events such as falls, fractures, and even death in affected individuals [1,2,3]. Skeletal muscle cells are unable to proliferate by division, and require muscle satellite cell (MuSC) differentiation to facilitate replenishment and repair of muscle fibers. This occurs via a series of closely interrelated steps, the most important of which is the transformation of myoblasts into myotubes [4,5], known as myogenic differentiation. As the differentiation ability of satellite cells decreases with age [6], the differentiation capacity of myoblasts is an indicator of skeletal muscle health. Myoblasts ensure the progression of myogenic differentiation by altering the cell cycle [7] and by expressing Myogenin and the myogenesis-related transcription factor MyoD [8]. Furthermore, an important event during myogenic differentiation is the change in cellular energy metabolism from glycolysis to oxidative phosphorylation [9]. Therefore, the prevailing doctrine states that the function and biosynthesis of mitochondria, as the primary organelle of cellular oxidative phosphorylation, are directly related to the success of myogenic differentiation [10,11,12]. Earlier studies have shown that the depletion of mitochondrial electron transport chain (ETC) complexes in myoblasts by inhibition of ETC function, such as by ethidium bromide, resulted in impaired differentiation of murine C2C12 myoblasts [13,14], suggesting that oxidative phosphorylation is essential for myogenic differentiation.

The key players in oxidative phosphorylation, the ETC complexes, are composed of subunits encoded by both nuclear DNA (nDNA) and mitochondrial DNA (mtDNA). It has been demonstrated that upregulation of mtDNA transcription factor A (TFAM) activates the transcription of mtDNA and can be beneficial for muscle regeneration [15]. In contrast, deficiency of mtDNA translation or mitonuclear protein imbalance leads to failed myogenic differentiation and the development of skeletal muscle degenerative diseases [16,17]. Existing evidence indicates that mitochondrial–nuclear crosstalk plays a regulatory role during myogenic differentiation, [18,19,20], suggesting that the study of mitochondria-to-nucleus retrograde signaling may provide new clues for the interpretation of myogenic differentiation.

The mitochondrial unfolded protein response (UPRmt) is a form of response caused by retrograde signaling from the mitochondria to the nucleus, aimed at maintaining the quality and functional integrity of the mitochondrial proteome [21,22]. UPRmt is activated under stressful conditions, such as impairment of mitochondrial function or imbalance of mitonuclear protein. This increases the expression of mitochondrial biosynthesis-related genes such as PPARG coactivator 1 alpha (PGC1a), nuclear respiratory factor 1(NRF1), and TFAM [23] as well as molecular chaperones and proteases such as the heat shock protein (HSP) family, lon peptidase 1 (LONP1), and High-temperature requirement protein A2 (HtrA2/Omi) [24,25,26].

One of the main triggers of the UPRmt is the disturbance of proteostasis in the mitochondrial intermembrane space (IMS) [24,27]. The IMS is important for the import and processing of nuclear-encoded proteins and the association of nDNA- and mtDNA-encoded ETC subunits on the inner mitochondrial membrane to form the ETC complex [28,29]. Under physiological conditions, proteins accumulated in the IMS are degraded by IMS proteases such as HtrA2, OMA1, and Yme1L [27,30,31]. However, stress or mitonuclear protein imbalance induces protein accumulation in the IMS, which in turn activates the UPRmt [32]. HtrA2/Omi is a protease localized to the IMS in the physiological state that is involved in the maintenance of mitochondrial homeostasis [33]. HtrA2 is considered to be an important protease involved in the UPRmt [24]. An earlier study showed that mice with motor neuron degeneration 2 (mnd2) harbor a HtrA2/Omi Ser276Cys missense mutation that results in the loss of HtrA2/Omi protease activity [34]. In another study by our group on skeletal muscle in mnd2 mice, we found a distinct phenotype of sarcopenia accompanied by an imbalance in mitonuclear expression [17]. Based on this, we hypothesize that HtrA2 may be involved in regulating myogenesis via retrograde signaling. We speculate that mitonuclear crosstalk is critical in determining myogenic differentiation, wherein HtrA2 may play a regulatory role in this process through UPRmt.

Herein, we propose the use of HtrA2 as an entry point to explore the possibility that IMS proteostasis acts as retrograde signaling to inhibit the differentiation of myogenic cells by interfering with the enzymatic activity of HtrA2. To model HtrA2 protease activity deficiency, we used mnd2 mice for in vivo experiments, as well as C2C12 myoblasts that were treated with UCF101, a specific inhibitor of HtrA2, for in vitro experiments. In order to discover how HtrA2 protease activity deficiency affects skeletal muscle tissue, we performed morphological and molecular experiments on gastrocnemius muscle tissue of mnd2 mice. We found that the deficiency of HtrA2 protease activity results in IMS proteostasis disorder and mitonuclear imbalance, induced UPRmt, and repressed myogenic differentiation in C2C12 myoblasts. In addition, the deficiency of HtrA2 protease activity led to the degeneration of skeletal muscle in mnd2 mice. In summary, we tested the hypothesis that the imbalance of mitochondrial IMS proteostasis acts via retrograde signaling to inhibit myogenic differentiation via UPRmt. These novel mechanistic insights may have implications for the development of new strategies for the treatment of sarcopenia.

## 2. Results

### 2.1. HtrA2/Omi Protease Deficiency Induces Sarcopenia in Mnd2 Mice

According to previous reports, the mitochondria in skeletal muscles and neuronal cells of mnd2 mice show conformational changes in mtDNA, elevated numbers of mistranslated and unfolded proteins, and significant impairment of mitochondrial structure and function [34]. In the current study, we performed HE staining on the cross-section area (CSA) of gastrocnemius muscle from 35-day-old wild-type (WT), HtrA2^mnd2(+/−)^, and HtrA2^mnd2(−/^^−)^ mice (Figure 1a–c). We found that HtrA2^mnd2−/−^ mice showed significant myofiber fragmentation and a larger myofiber gap when compared to WT mice. In addition, we performed semi-automatic analysis of HE staining images using ImageJ, for which 86 WT, 131 HtrA2^mnd2(+/−)^, and 123 HtrA2^mnd2(−/−)^ myofibers were used for area calculation and statistical analysis. As shown in Figure 1d, the majority of the CSAs of the HtrA2^mnd2(+/−)^ and HtrA2^mnd2(−/−)^ genotypes were less than 2000 μm^2^, whereas most of them were 2000 μm^2^ or more in WT. To investigate whether changes in the gastrocnemius muscle of mnd2 mice were related to the fiber type, we examined the transcriptional levels of myosin heavy chain (Myh) 1/2/4/7. The results showed that mRNA levels of Myh7 and Myh2 in HtrA2^(mnd2−/−)^ mice were significantly lower than in the WT, whereas those of Myh1 and Myh4 did not change significantly (Figure 1e). Furthermore, we found that expression of myosin and myogenin, which are markers of myogenic differentiation, was significantly reduced in the gastrocnemius tissues of HtrA2^mnd2(+/−)^ and HtrA2^mnd2(−/−)^ mice (Figure 1f). These results suggest that the skeletal muscles of HtrA2^mnd2(+/−)^ and HtrA2^mnd2(−/−)^ mice were significantly atrophied and constituted type I and type IIA fibers, as well as that mnd2 mice may have impaired muscle regeneration. Because HtrA2 is thought to play a role in mitochondrial proteostasis, we examined the expression levels of mitochondrial proteins in the gastrocnemius muscle. We found that RNA polymerase mitochondrial (POLRmt), TFAM, and transcription factor B2 mitochondrial (TFB2M) levels were reduced in HtrA2^mnd2(+/−)^ and HtrA2^mnd2(−/−)^ mice (Figure 1g). At the same time, the expression of the IMS proteases S-OMA1, the active form of OMA1 [35], as well as Yme1L1, were increased (Figure 1h). This suggests that gastrocnemius muscle atrophy in HtrA2^mnd2−/−^ and HtrA2^mnd2−/−^ mice may be associated with disturbance of mitochondrial protein homeostasis and dysfunction of mtDNA transcription.

### 2.2. HtrA2/Omi Protease Deficiency Impairs the Differentiation of C2C12 Myoblast

Based on the above results, we theorized that the IMS protease HtrA2 may play a key role in mitochondrial gene expression and myogenic cell differentiation. To test this hypothesis, we treated murine C2C12 myoblasts with UCF101, a specific inhibitor of HtrA2, and performed myogenic differentiation. The molecular structure of UCF101 is shown in Figure 2a; in previous studies, it has been reported to have a significant inhibitory effect on HtrA2 protease activity at a concentration of 20 μM [36,37,38]. C2C12 myoblasts were treated with different concentrations of UCF101 for 48 h. The results showed that cell viability was not affected when the concentration of UCF101 was below 20 μM (Figure 2b). We divided the cells into groups that were either treated with 20 μM of UCF101 or were untreated, representing the HtrA2-protease-deficient group and normal differentiation group, respectively. Upon observation of differentiation (Figure 2c), we found that myotubes were visible in the untreated group under 40× magnification on day 3 (D3) and that their number increased by D5. In contrast, the number of myotubular cells as well as the length and diameter of the myotubes was significantly lower in the UCF101-treated group. Subsequently, we measured the expression level of myosin, a marker of myogenic differentiation, by immunofluorescence staining of myosin protein and of the nuclei on D3 and D5 of differentiation, which was followed by calculation of the myosin fusion index (FI) (Figure 2d–e). We found that the myosin fluorescence intensity and area of the treated group were lower than those of the untreated group. The FI in the treated group was 11.59% and 23.57% at D3 and D5, respectively, which was significantly lower than the FI in the untreated group (16.48% and 41.13%, respectively). The results from the Western blot of Myogenin and Myosin demonstrate that differentiation in the C2C12 myoblasts was repressed in the UCF101-treated group (Figure 3a). A similar trend was observed in the levels of Myh1/2/4/7 mRNA transcripts on D3 and D5 (Figure 3b). These results indicate that HtrA2 protease deficiency represses myogenic differentiation at both the transcriptional and the translational levels. Significant repression of myogenic differentiation resulted in significant impairment of myotubular cell generation.

### 2.3. HtrA2/Omi Protease Deficiency Causes OXPHOS Hypofunction

To verify whether the impairment of myogenic differentiation was related to the pro-apoptotic effect of HtrA2, we performed flow-cytometry-based assays to measure apoptosis and reactive oxygen species (ROS) levels in C2C12 myoblasts on D1, D3, and D5 (Figure 4a,e). We found no significant changes in the extent of apoptosis or ROS levels between the two groups during the whole experiment. Therefore, we concluded that the delayed differentiation of C2C12 myoblasts caused by UCF101 did not directly correlate with the pro-apoptotic effect of HtrA2. To assess oxidative respiratory function and mitochondrial function in C2C12 myoblasts during differentiation, we examined the ATP content, 24 h glucose uptake, and 24 h lactate secretion of D1, D3, and D5 cells (Figure 4b–d). We found that while the 24-h glucose uptake gradually increased with myogenic differentiation, there was no significant difference between the untreated and treated groups. However, we found that lactate secretion in the UCF101 treated group was significantly higher than in the untreated group, indicating that glycolysis was the major mode of energy metabolism in the treated group. In addition, the ATP content at D3 and D5 in the UCF101-treated group was lower compared to the untreated group. Therefore, we suggest that HtrA2 protease deficiency prevents the shift of energy metabolism to oxidative phosphorylation in C2C12 myoblasts during myogenic differentiation. 

### 2.4. HtrA2/Omi Protease Deficiency Leads to Mitonuclear Imbalance

To further investigate the reason for the reduction in oxidative phosphorylation in C2C12 myoblasts in the absence of HtrA2 protease activity, we measured the expression of the subunits of the ETC complex. For this, we performed western blot assays on D1, D3, and D5 for the nDNA-encoded proteins cytochrome c oxidase subunit 4 (COXIV) and succinate dehydrogenase complex flavoprotein subunit A (SDHA), as well as for the mtDNA-encoded proteins NADH de-hydrogenase subunit 1 (ND1), cytochrome b (CYTB), and ATP synthase F0 subunit 6 (ATP6A1). The expression of COXIV and SDHA did not differ significantly, whereas the expression of mtDNA-encoded ETC subunits was significantly lower in the treated group than that in the untreated group (Figure 5a). In addition, we examined mitochondrial biogenesis via the PGC1α-NRF1/2 signaling pathway (Figure 5b) and found that the expression of NRF2 in the UCF101-treated group was lower than in the untreated group. This is consistent with our previously proposed theory that the mtDNA-encoded subunits affect in mnd2 mice are regulated by NRF2 [17]. To investigate how this mitonuclear protein imbalance occurred, we first examined the number of mitochondria on D1, D3, and D5 using flow cytometry. When compared to the green fluorescence intensity observed on D1 in the UCF101-treated group, a leftward shift of 12.04% and 23.44% was observed on D3 and D5, respectively (Figure 6a). This was lower than the fluorescence intensity observed in the untreated group during the same period. At the same time, the amounts of mitochondrial DNA-binding proteins TFAM, TFB2M, and POLRmt as well as the mitochondrial copy number assay (with ND1, COXII, and CYTB as mtDNA marker genes) showed that the mtDNA copy number gradually increased, approximately doubling during differentiation from D1 to D5 in the untreated group. In contrast, this value increased less than 1.5-fold in the treated group. In addition, the expression levels of TFAM, TFB2M, and POLRmt were lower in the treated group when compared to the untreated group (Figure 6b,c). These results reveal that deficiency of HtrA2 protease activity inhibits mitochondrial biogenesis during myogenic differentiation and leads to a reduction in mtDNA copy number and loss of mitochondrial DNA-binding proteins. As mitochondrial DNA contains 37 genes, including 13 subunits of the mitochondrial respiratory chain complex, a reduction in copy number may result in the loss of mtDNA-encoded ETC complex units. These results demonstrate that a mitonuclear protein imbalance occurs during myogenic differentiation when the protease activity of HtrA2 is inhibited. This imbalance in the ETC complex subunits is an important reason for deficient oxidative phosphorylation and impaired myogenic differentiation. 

### 2.5. HtrA2/Omi Protease Deficiency Activates UPRmt during Myogenic Differentiation

Our observation of mitonuclear protein imbalance in C2C12 myoblasts during myogenic differentiation prompted our investigation into the role of the UPRmt in this process. We measured the expression level of mitochondrial proteases and the molecular chaperones involved in UPRmt on D1, D3, and D5 during myogenic differentiation. The results (Figure 7a,b) showed that the expression of IMS proteases Yme1L1 and both L- and S-OMA1 in the UCF101 treated group gradually increased after the initiation of differentiation, and the protein amounts were significantly higher than those in the untreated group. The expression of HtrA2 protein increased in treated group on D1 and decreased at D3. This phenomenon may be related to inhibition of the HtrA2 protease activity. Thus, UCF101 led to impairment of IMS proteostasis during myogenic differentiation. The Western blot and quantitative real-time PCR (qPCR) assays show that the expression levels of lon peptidase 1 (LONP1), heat shock proteins 10, 60, and 75 (HSP10, 60, and 75), and other molecules involved in UPRmt were significantly elevated in the treated group on D3 and D5 when compared to the untreated group during the same period (Figure 7c). The mRNA transcription levels of the aseinolytic mitochondrial matrix peptidase proteolytic subunit (Clpp), the caseinolytic mitochondrial matrix peptidase chaperone subunit X (Clpx), LonP1, and the molecular chaperones Hsp10/60/75 were higher on D3 in the treated group than in the untreated group. To verify the upstream signal of UPRmt, we examined the protein levels of DNA-damage-inducible transcript 3 (CHOP), eukaryotic translation initiation factor 2 subunit alpha (eIF2α), and p-eIF2α, and found that the level of p-eIF2α was lower in the untreated group at the initial stage of differentiation. Meanwhile the levels of CHOP and p-eIF2α were much higher in the treated group, whereas the expression of eIF2α was not changed. These results suggest that mitochondrial proteostasis is disturbed and UPRmt is activated during myogenic differentiation following HtrA2 protease inhibition.

## 3. Discussion

Inhibition or stimulation of mitochondrial activity is known to respectively abolish or promote myoblast differentiation through various pathways [11,12]. In this study, we investigated the effect of mitochondrial IMS proteostasis on myogenic differentiation and the role of UPRmt by treating C2C12 cells with UCF101, a specific inhibitor of HtrA2 enzyme activity.

Initially, we performed in vivo experiments using mnd2 mice, which are a model of mitochondrial protease HtrA2/Omi Ser276Cys mistranslation mutation characterized by neurodegenerative lesions, motor dysfunction, and premature death [34]. By analyzing cross-sections of gastrocnemius muscle fibers from mnd2 mice, we confirmed that depletion of HtrA2 enzyme activity leads to atrophy of skeletal muscles. Furthermore, the mRNA transcript levels of Myh2 and Myh7 were decreased in these tissues. These myosin heavy chain isoforms are marker genes for type I and IIA muscle fibers, and their main mode of energy metabolism is based on oxidative phosphorylation [39]. This suggests that the absence of HtrA2 enzyme activity affects the functioning of oxidative phosphorylation in the mitochondria of myofibrils. We found significantly lower expression of myogenin and myosin in heterozygotes and purets than in the wild type. Both of these marker proteins are associated with muscle differentiation, with myogenin thought to be required for muscle production during late embryonic development in mice [40]. These results imply that the cause of skeletal muscle atrophy in mnd2 mice may be the impairment of myogenic differentiation during development. 

To further investigate the cause of sarcopenia due to deficiency of HtrA2 activity, we used UCF101 to inhibit HtrA2 enzymatic activity during myogenic differentiation of C2C12 cells. We found a repressive effect of UCF101 on C2C12 myogenic differentiation both microscopically and at the molecular level. UCF101 has been used in numerous studies for its ability to significantly inhibit the protease activity of HtrA2, thus preventing the binding of HtrA2 to proteins involved in unfolded protein reactions and pyruvate metabolism in mitochondria [36,37,41]. In our experiments, we found no significant change in the level of apoptosis in the UCF101 treated and untreated groups. Therefore, we concluded that the inhibitory effect of UCF101 on myogenic differentiation did not originate from its inhibitory effect on apoptosis. Klupsch et al. found that UCF101 activates CHOP transcription and UPR, although a mechanism for UCF101 mediated UPR activation was not proposed [38]. Similarly, in our experiments we observed an increase in CHOP expression and activation of the UPRmt, which we attribute to the inhibition of HtrA2 enzymatic activity in mitochondrial IMS. We found that deficiency in HtrA2 activity led to elevated expression of the mitochondrial IMS proteases OMA1 and YME1L1, which have recently been shown to (i) transmit stress signals into cells by cleaving DAP3 binding cell death enhancer 1(DELE1) and (ii) coordinate with TIM23 for regulation of mitochondrial protein import [42,43]. These changes in IMS proteases were observed in mnd2 mice and during C2C12 myogenic differentiation. Taken together, these results suggest that the depletion of IMS HtrA2 enzymatic activity by UCF101 triggers an imbalance in IMS proteostasis, and that this imbalance is the underlying cause of inhibition of myogenic differentiation.

It is generally accepted that during differentiation of stem cells the mode of cellular energy metabolism shifts from glycolysis to oxidative phosphorylation [44,45,46,47,48], and that impairment of oxidative phosphorylation leads to impaired myogenic differentiation [16,17]. In this study, oxidative phosphorylation was significantly lower in UCF101-treated C2C12 cells than in the untreated group, and Western blot results indicated that this was due to a mismatch between the mtDNA- and nDNA-encoded ETC complex subunits. This phenomenon is consistent with our team’s earlier findings in mnd2 mice, in which we observed decreased expression of ND1, CYTB, and ATP6A1 [17]. These ETC complex subunits are only encoded by mtDNA, the content of which must be adjusted according to metabolic needs [49]. In that study, we observed a decrease in the level of mtDNA copy number, which could explain the loss of mtDNA-encoded ETC complex subunits. In addition, in the present study, we found that POLRmt, TFAM, and TFB2M expression level was significantly reduced in mnd2 mice. As these three proteins are directly associated with mitochondrial DNA stabilization and transcription initiation [50], this explains, at least to a certain extent, the significant decrease in mitonuclear phosphorylation in mnd2 mice and the mitonuclear protein imbalance observed in mnd2 mice and C2C12 myoblasts.

Both mitonuclear protein imbalance and mitochondrial IMS stress are major triggers of UPRmt [23,24,25,26]. Therefore, we theorized that the UPRmt may be correlated with the inhibition of C2C12 myoblast differentiation. Through Western blotting, we observed increased expression of CHOP, HSP-family molecular chaperones, and LONP1, suggesting that depletion of HtrA2 function activates the UPRmt during myogenic differentiation. A widely accepted hypothesis is that the depletion of HtrA2 activity impairs mitochondrial function and activates the UPRmt by disrupting mitochondrial proteostasis and increasing ROS production [24,51,52,53]. However, we did not find any upregulation of UPRmt-related molecules in mnd2 mice in our earlier study [17], which we believe may be related to the different physiological state of myoblasts during differentiation as compared to mature skeletal muscle cells. We intend to verify this in future studies.

Currently, no unified theory exists regarding the relationship between the UPRmt and myogenesis, although it has been reported that the deletion of Clpp and LONP1 leads to impaired differentiation of myogenic cells [54,55]. It has been shown that overexpression of CHOP or HSP family proteins during early differentiation hinders the differentiation of myogenic cells into myotubular cells [56,57]. In the present study, we suggest that activation of UPRmt early in the differentiation process is a detrimental signal for cellular differentiation, although UPRmt itself is a cellular rescuer of mitochondrial function. Activation of CHOP and p-eIF2α early in differentiation may lead to inhibition of MyoD translation [56] and of overall translation of cellular proteins, respectively. In addition, their activation inhibits the process of myogenic differentiation.

In conclusion, we found that the depletion of the enzymatic activity of HtrA2/Omi leads to degenerative lesions and imbalance of mitochondrial proteostasis in murine skeletal muscle. This was verified by experiments with C2C12 myogenic cells, which revealed that this enzymatic activity leads to blocked differentiation of myogenic cells. In our previous study, we found that loss of HtrA2 enzymatic activity leads to mitochondrial–nuclear protein imbalance in mouse skeletal muscle tissue. In the present study, we found that this imbalance occurs during the differentiation of adult myoblasts and is accompanied by a decrease in the amounts of mtDNA and mtDNA transcription complex protein during differentiation. This leads to a decrease in oxidative phosphorylation and the number of the mitochondrial genes encoding the ETC subunit, which in turn activates the UPRmt during the differentiation of adult myoblasts, accompanied by an increase in eIF2α phosphorylation and CHOP expression. Translational inhibition of cell-wide proteins by a premature UPR is a possible cause of this delayed differentiation (Figure 8). The results of this study may have implications for the development of new strategies for the treatment of sarcopenia.

## 4. Materials and Methods

### 4.1. Animals and Genotyping

The heterozygous mnd2 (HtrA2^mnd2(+/−)^) mice (Stock Number: 004608), SPF grade, were obtained from the Jackson Laboratory (Bar Harbor, ME, USA). HtrA2^(mnd2+/−)^ mice were interbred to generate populations having all three genotypes (wild-type, heterozygous, and homozygous). Mice were kept at a constant temperature (22 °C) with a light/dark cycle of 12 h and were provided access to food and tap water. Gastrocnemius muscles from thirty-five-day-old male mice for each genotype (*n* = 6) were used in this study. All animals used in the procedures were handled according to the Guide for the Care and Use of Laboratory Animals, and the in vivo experimental methods were approved by the University Committee on the Use of Animals of Jilin University, China (Research and Examination No. 1982021; approval on 22 June 2021), laboratory animal use license number (SYXK(JI)2018-0001).

Experimental offspring were identified by PCR restriction enzyme analysis according to the protocol provided by the Jackson Laboratory. Briefly, the Ser276Cys mutant allele of HtrA2 was detected by amplification of a 500 base pair (bp) fragment containing exon-3 from nDNA using the mnd2 primers (Table 1). Digestion of genomic DNA with AluI produced a 244 bp product from the homozygous mice, compared to 171 and 73 bp fragments from the wild type mice. PCR assays of the digestion fragments were performed, with the products separated on agarose gel and stained with M5 Gelred-Plus (Mei5bio, Beijing, China).

### 4.2. Histological and Morphological Quantitative Analysis of Muscle Fibers 

Gastrocnemius muscles were fixed with 4% paraformaldehyde, embedded in paraffin, dehydrated, and rehydrated before 4μm coronal and sagittal sections were stained with hematoxylin and eosin (HE). Finally, we observed the morphological characteristics of the muscles using a microscope (ECLIPSE Ci-L; Nikon, Japan). Quantitative morphological analysis of the muscle fibers was performed using ImageJ software (Version 1.52a http://rsb.info.nih.gov/ij/, accessed on 3 August 2022) using images of HE-stained cross sections. Briefly, Open-CSAM, an ImageJ macro supporting quantitative analysis of muscle fibers, was added to ImageJ according to the work of Thibaut et al. [58]. HE images were converted to grayscale in 8-bit with Photoshop software (Adobe Inc., San Jose, CA, USA) before being imported into Open-CSAM. Finally, the parameters of the muscle fibers analyzed using Open-CSAM were exported.

### 4.3. C2C12 Myoblast Culture and Differentiation

Mouse C2C12 myoblasts were purchased from the cell bank of the Chinese Academy of Sciences (Shanghai, China). For maintenance of C2C12 myoblasts, cells were cultured in normal growth medium consisting of Dulbecco’s modified eagle medium (DMEM) supplemented with 10% Foetal Bovine Serum (FBS), 100 U/mL penicillin, and 100 μg/mL streptomycin in 5% CO_2_ at 37 °C. The growth medium was changed out every day.

For myogenic differentiation, C2C12 myoblasts at 70–80% confluence were transferred to differentiation medium (DM) with 2% horse serum. One group of these cells were treated with UCF101, while the other group was left untreated to serve as a control. The medium for both groups was replaced daily with fresh DM with or without UCF101, as appropriate.

### 4.4. Measurement of Cell Viability

Cell viability was determined via MTT assay. Cells were seeded in 96-well plates and incubated in culture medium until 70–80% confluency. The cells were treated with 0, 1.25, 2.5, 5, 10, 20, or 40 μM UCF101 for 48 h. Next, the cells were incubated in the dark with MTT reagent (0.5 mg/mL) at 37 °C for 2 h. Following this, the medium was removed, formazan was dissolved in DMSO, and the absorbance at 540 nm was measured using a microplate reader (Thermo Fisher Scientific, Vantaa, Finland).

### 4.5. Immunofluorescence Microscopy

C2C12 myoblasts were seeded into 24-well plates and treated with UCF101 or were left untreated. It has been suggested through many in vitro and in vivo studies that UCF-101 can specifically inhibit HtrA2 protease activity at a concentration of 20 μM [36,37,38]. On D3 and D5, cells were fixed in 4% paraformaldehyde for 15 min and permeabilized with 0.1% Triton X-100 for 7 min. Cells were blocked with 5% bovine serum albumin at room temperature (RT) for 30 min and subsequently incubated with primary antibody Myosin (MF20, DSHB) overnight at 4 °C. Cells were incubated at RT with fluorescein isothiocyanate (FITC)/Texas Red-conjugated secondary antibodies (Proteintech, Chicago, IL, USA) for 0.5 h and with Hoechst 33,342 for 5 min. The images were acquired using an Echo Lab Revolve microscope (San Diego, CA, USA). The myotube fusion index (FI) was calculated as the percentage of total nuclei incorporated in myotubes.

### 4.6. Quantitative Real-Time PCR 

Total RNA from C2C12 myoblasts was extracted using TRIzol reagent (Invitrogen, Carlsbad, CA, USA). cDNA was synthesized from the RNA products using the Omniscript Reverse Transcription Kit (QIAGEN, Hilden, Germany). RT-qPCR was performed to estimate mRNA levels using SYBR SuperMix (TransGen Biotech, Beijing, China). All reactions were performed twice using a Bio-Rad CFX96 Touch Real-Time PCR System (Bio-Rad, Hercules, CA, USA). Glyceraldehyde-3-phosphate dehydrogenase (GAPDH) was used as an internal control. The relative expression level for each gene was calculated using the 2^−ΔΔCt^ relative quantification method and normalized to the endogenous level. The primer sequences used are listed in Table 2.

### 4.7. Western Blot Analysis

C2C12 myoblasts were seeded into 6-well plates and UCF101 was added to the treatment group following initiation of differentiation. Cells were harvested at D1, 3, 5, and GM. Cells were pulverized and lysed with 200 μL of RIPA buffer (Beyotime, Beijing, China). Lysates were ultrasonicated for 6 × 3 s on ice and placed on ice for 45 min. Next, the lysates were centrifuged at 4500× *g* for 15 min at 4 °C and the precipitate was discarded. Protein concentrations in the supernatants were determined using Bradford reagent (Bio-Rad, Hercules, CA, USA). The protein samples (10 μg) were resolved by 10–12% sodium dodecyl sulfate–polyacrylamide gel electrophoresis and transferred onto polyvinylidene fluoride (PVDF) membranes (Millipore, Billerica, MA, USA). Finally, immunodetection was performed using an enhanced chemiluminescence detection kit (DW101, TransGen Biotech, Beijing, China) and images were captured using Syngene Bio Imaging (Synoptics, Cambridge, UK). The following primary antibodies were used: heat-shock protein 75 (HSP75, 10325-1-AP), HSP60 (15282-1-AP), lon peptidase 1 (LONP1, 15440-1-AP), C-EBP homologous protein (CHOP, 15204-1-AP), Oma1 (17116-1-AP,), ND1 (19703-1-AP), CYTB (55090-1-AP), COXIV (11242-1-AP), SDHA (14865-1-AP), ATP6A1 (17115-1-AP), PGC1α (66369-1-Ig), NRF1 (12482-1-AP), TFAM (22586-1-AP), TFB2M (22441-1-AP), antibodies were purchased from Proteintech (Chicago, IL, USA). GAPDH (AC002), POLRmt (A15605), NRF2/GAPBA (A8419), antibodies were purchased from ABclonal (Abclonal, Boston, MA, USA). HSP10 (sc-376313), VDAC1 (sc-390996), antibodies were purchased from SantaCruz (Dallas, TX, USA). Phospho-eIF2α (Ser51) (AF3087), eIF2α (AF6087), antibodies were purchased from Affinity (Cincinnati, OH, USA). Myogenin (F5D, DSHB), Myosin (MF20, DSHB), antibodies were purchased from DSHB (Iwoa, IA, USA). All antibodies were used at a 1:1000 dilution.

### 4.8. Quantification of Mitochondrial Copy Numbers

Mitochondrial copy numbers were measured using absolute RT-qPCR, as previously described [59]. Briefly, genomic DNA from C2C12 myoblasts was isolated using the DNeasy Blood and Tissue Kit (QIAGEN, Hilden, Germany) according to the manufacturer’s instructions. Quantification of mtDNA copy number was performed in triplicate by qRT-PCR. mNADH1, mCOXII, and mCYTB were used as mtDNA markers and the β-globin nuclear intron was used as the nDNA marker. Relative gene expression was normalized to that of the β-globin gene (ΔCT) in each sample and to the GM. The primer sequences used are listed in Table 3.

### 4.9. Analysis of Glucose Uptake and Lactate Secretion

To determine the effects of UCF101 on glucose metabolism in C2C12 myoblasts during differentiation, glucose uptake capacity and lactate secretion were analyzed using a glucose assay kit and a lactic acid assay kit, respectively (both kits produced by Nanjing Jiancheng Bioengineering Institute, Nanjing, China). The assays were performed according to the manufacturers’ instructions. Briefly, UCF101 was added to the treatment group of C2C12 cells at the beginning of the differentiation experiment. At D1, 3, and 5, the DM of the treated cells was collected and centrifuged at 3000× *g* for 5 min to remove debris. After centrifugation, the supernatants were mixed with the working solution in a 96-well plate. Following 5 min of incubation at 37 °C, absorbance was measured at 505 nm. Because we monitored the residual glucose content in the culture medium, glucose consumption in each assay was calculated as the glucose content in fresh culture medium minus the residual glucose content in the culture medium at the conclusion of the experiment. Total cellular protein was used to normalize the glucose consumption. The lactic acid test was preheated at 37 °C for 10 min, followed by the addition of stop solution. Absorbance values were measured at 530 nm, and the relative uptake and secretion were normalized to protein abundance.

### 4.10. Measurement of ATP Content

ATP content in the C2C12 myoblasts was measured using a bioluminescent assay kit (Beyotime, Shanghai, China) according to the manufacturer’s instructions. Briefly, fresh cell lysates were collected and then centrifuged at 12,000 *g* at 4 °C for 10 min. The supernatant was added to the detection reagent. ATP content was measured using a multimode microplate reader (FLUOstar Omega, BMG Labtech, Ortenberg, Germany), with the relative luminescence units normalized to the protein abundance.

### 4.11. Flow Cytometry

C2C12 myoblasts were seeded in 6-well plates and treated with UCF101 or DMSO after differentiation was initiated. The cells were harvested at D1, 3, and 5 and stained according to the manufacturer’s instructions. Apoptosis was evaluated using an Annexin V-FITC PI double staining kit (BD Biosciences, San Diego, CA, USA). The number of mitochondria was measured using a MitoTracker Green probe (Beyotime, Shanghai, China). ROS levels were measured using a Reactive Oxygen Species Assay Kit (Beyotime, Shanghai, China). Samples were analyzed by flow cytometry (Guava easyCyte, Austin, TX, USA).

## Figures and Tables

**Figure 1 ijms-23-11761-f001:**
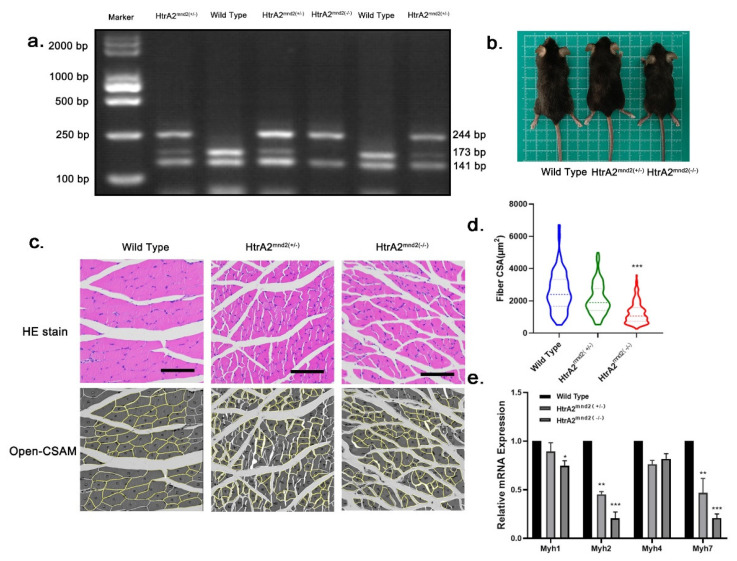
Gastrocnemius muscle tissue from HtrA2^mnd2(−/−)^ mice exhibiting a sarcopenic phenotype. (**a**) Agarose gel electrophoresis graphs of genotype identification, fluorescence of DNA fragments under UV for the three genotypes. (**b**) Thirty-five-day-old mice corresponding to genotype WT (left), HtrA2^mnd2(+/−)^ (middle), and HtrA2^mnd2(−/−^) (right). (**c**) HE staining of cross-sections of gastrocnemius muscle tissues from mice with the three genotypes and high-throughput semiautomatic muscle fiber CSA analysis by Open-CSAM. A total of 86 WT, 131 HtrA2^mnd2(+/−)^, and 123 HtrA2^mnd2(−/−)^ muscle fibers were identified and analyzed. (**d**) The CSA of HtrA2^mnd2(+/−)^ and HtrA2^mnd2(−/−)^ myofibers was significantly smaller compared to WT group mice. (**e**) RT-qPCR detection of mRNA transcript levels of Myh1, 2, 4, and 7 in gastrocnemius muscle of three genotypes of mice (*n* = 6). (**f**) Western Blot assay of expression of Myosin and Myogenin, marker proteins of myogenic differentiation, and quantification of western blots in gastrocnemius tissue of mice of the three genotypes (*n* = 6). (**g**) Western Blot assay of mitochondrial DNA binding proteins POLRmt, TFAM, and TFB2M expression and quantification of western blots in gastrocnemius muscle tissues of three genotypes (*n* = 6). (**h**) Western Blot detection of IMS proteases HtrA2, YME1L1, and OMA1 expression and quantification of western blots in gastrocnemius muscle tissues of three genotypes (*n* = 6). Two-tailed unpaired Student’s *t*-tests were used. Statistical significance: * *p* ≤ 0.05, ** *p* ≤ 0.01, *** *p* ≤ 0.001. Scale bar: 50 μm. CSA, cross-sectional area; WT, wild type; RT-qPCR, real time quantitative PCR; HE, Haematoxyline and eosin stain; Myh, myosin heavy chain; POLRmt, RNA polymerase mitochondrial; TFAM, transcription factor A, mitochondrial; TFB2M, transcription factor B2, mitochondrial; HtrA2, high temperature requirement protein A2; OMA1, OMA1 zinc metallopeptidase; YME1L1, i-AAA protease YME1 L1; mnd2, motor neuron degeneration 2 mutant.

**Figure 2 ijms-23-11761-f002:**
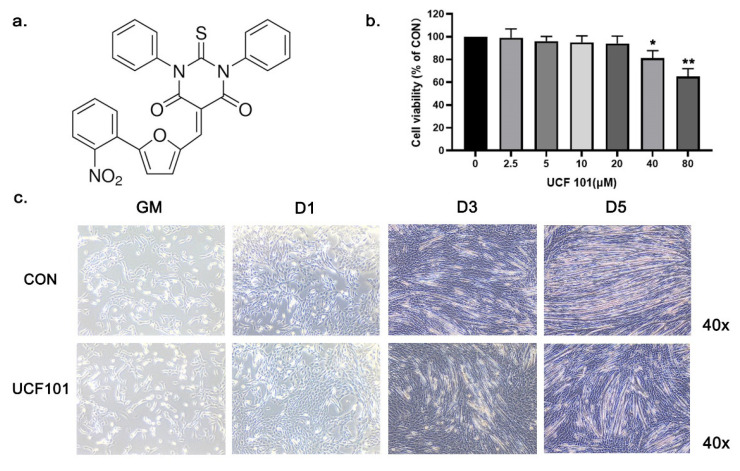
HtrA2/Omi protease deficiency impaired the differentiation of C2C12 myoblasts. (**a**) The chemical structure of UCF101. (**b**) Cells were treated with UCF101 for 48 h. The cell viability of C2C12 myoblasts was estimated by MTT assay. (**c**) Differentiation of C2C12 myoblasts treated with UCF101 (20 μM) or untreated on GM, D1, D3, and D5 at 40× magnification. (**d**) C2C12 myoblasts were treated with UCF101 (20 μM) or were untreated for 3 days and 5 days after differentiation. Immunofluorescence staining of Myosin (red) and the nucleus (blue) in C2C12 myoblasts (scale bar = 30 μm). (**e**) The FI of C2C12 myoblasts treated with UCF101 (20 μM) or untreated for 3 days and 5 days after differentiation. Fusion index (FI) was calculated as the percentage of total nuclei incorporated in myotubes. Two-tailed unpaired Student’s *t*-tests were used. Statistical significance: * *p* ≤ 0.05, ** *p* ≤ 0.01.

**Figure 3 ijms-23-11761-f003:**
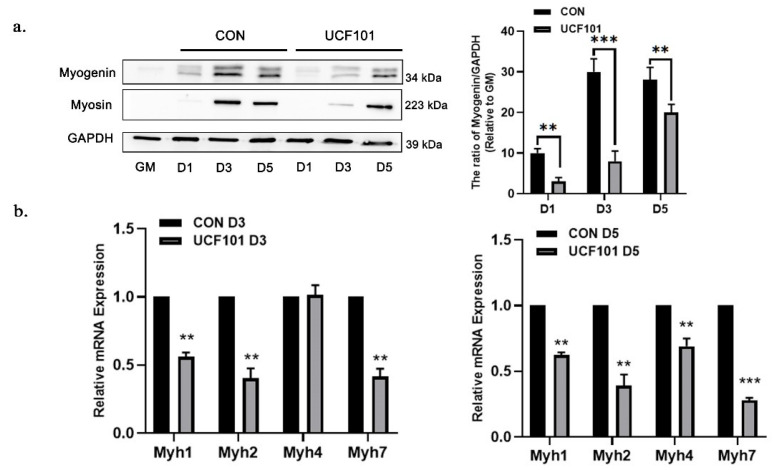
HtrA2/Omi protease deficiency reduced transcription and expression of differentiation marker proteins in C2C12 myoblasts. (**a**) Western blot of muscle differentiation marker proteins Myosin and Myogenin at differentiation on D1, D3, and D5 in C2C12 myoblasts treated and untreated with UCF101 (20 μM) and quantification of Western blots relative to GM (*n* = 3). (**b**) qPCR was performed to detect the transcript levels of Myh1, 2, 4, and 7 at differentiation on D1, D3, and D5 in C2C12 myoblasts treated and untreated with UCF101 (20 μM). Two-tailed unpaired Student’s *t* tests were used. Statistical significance: ** *p* ≤ 0.01, *** *p* ≤ 0.001.

**Figure 4 ijms-23-11761-f004:**
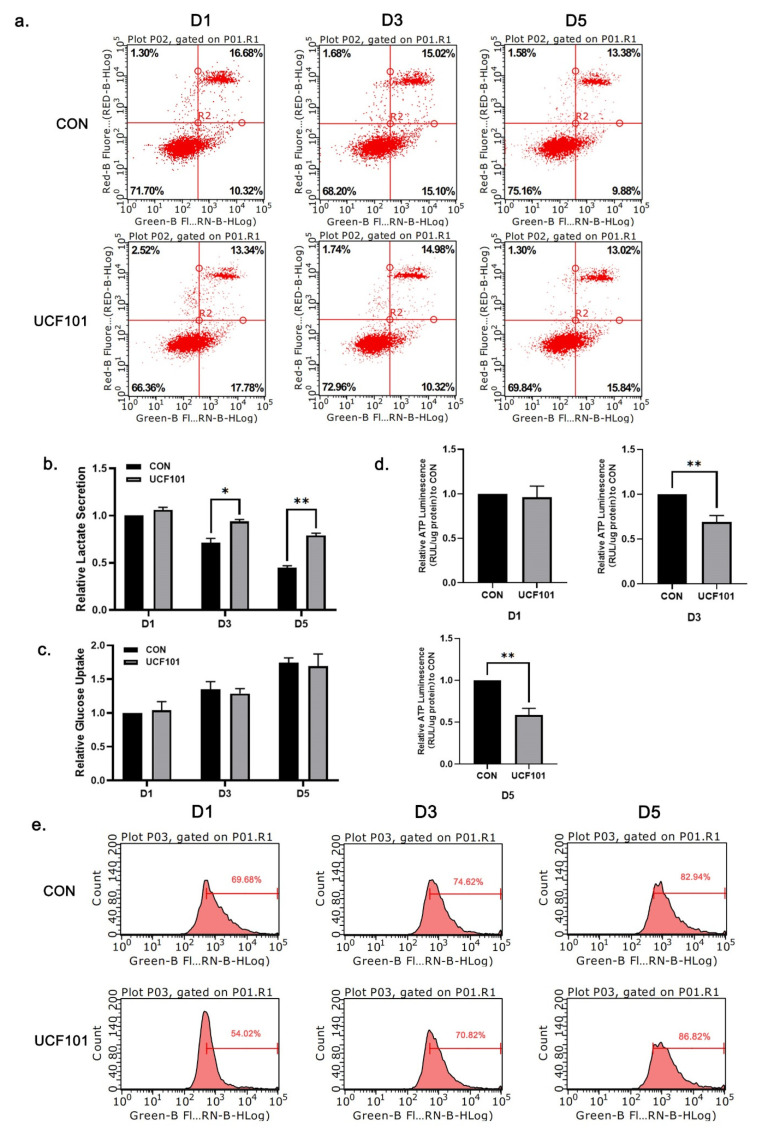
The levels of apoptosis, ROS, and energy metabolism markers during differentiation of C2C12 myoblasts. (**a**) Flow cytometric analysis of apoptotic rates of C2C12 myoblasts treated with UCF101 (20 μM) or untreated following differentiation on D1, D3, and D5. (**b**) Accumulation of lactate in C2C12 myoblasts treated with UCF101 (20 μM) or untreated since differentiation on D1, D3, and D5 within 24 h. Data were normalized to protein abundance. (**c**) Consumption of glucose of C2C12 myoblasts within 24 h. Data were normalized to protein abundance. (**d**) ATP content of C2C12 myoblasts determined using an ATP assay kit. RLU was normalized to protein abundance. (**e**) Flow cytometric analysis of ROS levels of C2C12 myoblasts treated with UCF101 (20 μM) or untreated following differentiation on D1, D3, and D5. Two-tailed unpaired Student’s *t*-tests were used. Statistical significance: * *p* ≤ 0.05, ** *p* ≤ 0.01; RLU, relative luminescence unit; ROS, reactive oxygen species.

**Figure 5 ijms-23-11761-f005:**
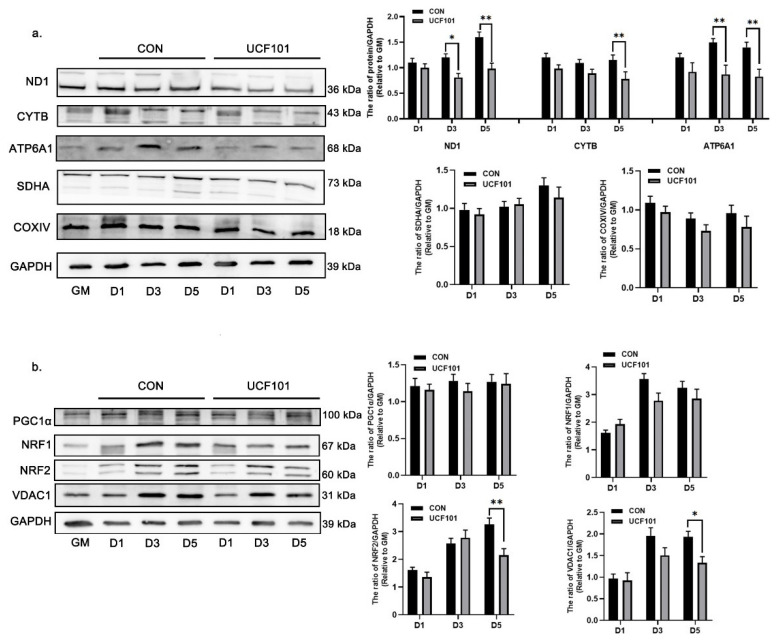
Expression levels of ETC complex subunits and PGC1-NRF1/2 signaling pathway proteins during differentiation of C2C12 myoblasts. (**a**) Western blot of ETC complex subunits ND1, CYTB, ATP6A1, SDHA, and COXIV in C2C12 cells during differentiation on D1, D3, and D5 after treatment with UCF101 (20 μM) or no treatment and quantification of Western blots relative to GM (*n* = 3). (**b**) Western blot of PGC1α, NRF1, NRF2, and VDAC in C2C12 myoblasts at differentiation on D1, D3, and D5 after treatment with UCF101 (20 μM) or no treatment and quantification of Western blots relative to GM (*n* = 3). Two-tailed unpaired Student’s *t*-tests were used. Statistical significance: * *p* ≤ 0.05, ** *p* ≤ 0.01. ND1, NADH1; SDHA, succinate dehydrogenase complex flavoprotein subunit A; COXIV, cytochrome c oxidase subunit 4; CYTB, cytochrome b; ATP6, ATP synthase F0 subunit 6.

**Figure 6 ijms-23-11761-f006:**
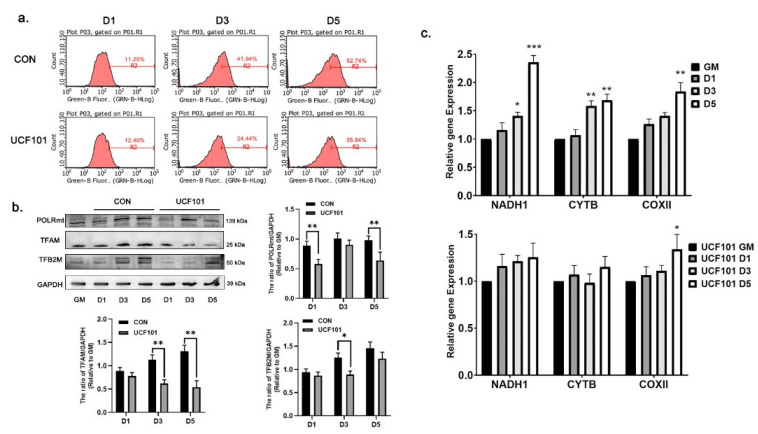
The mitochondrial number, mtDNA copy number, and mtDNA transcription complex levels during differentiation of C2C12 myoblasts. (**a**) The total number of mitochondria in C2C12 myoblasts treated with UCF101 (20 μM) or untreated following D1, D3, and D5 as measured by flow cytometry. (**b**) Western blot of mitochondrial DNA binding protein, POLRmt, TFAM, and TFB2M on D1, D3, and D5 and quantification of western blots relative to GM (*n* = 3). (**c**) qPCR assay of NADH1, CYTB, and ATP6 during myogenic differentiation, showing the relative mtDNA copy number change (*n* = 4), normalized to the copy number of β-globin. Two-tailed unpaired Student’s *t*-tests were used. Statistical significance: * *p* ≤ 0.05, ** *p* ≤ 0.01, *** *p* ≤ 0.001. CYTB, cytochrome b; ATP6, ATP synthase F0 subunit 6.

**Figure 7 ijms-23-11761-f007:**
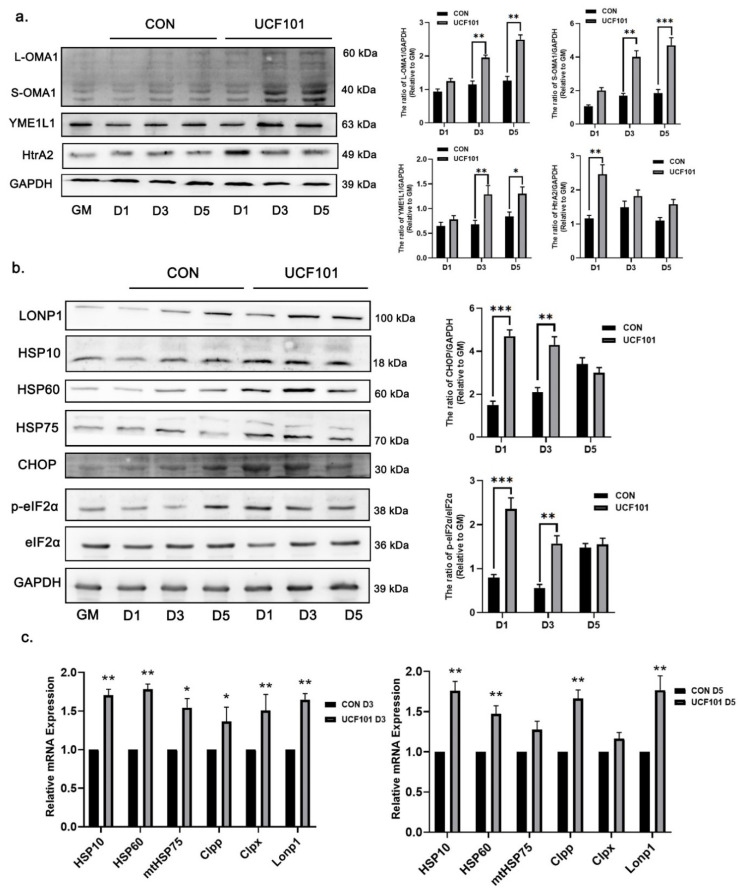
HtrA2/Omi protease deficiency activates UPRmt signaling during myogenic differentiation. (**a**) Western blot of IMS proteases OMA1, YME1L1, and HtrA2 on D1, D3, and D5 treated with UCF101 (20 μM) or untreated and quantification of western blots relative to GM (*n* = 3). (**b,c**) Western blot of protein expression and mRNA transcript levels of the UPRmt upstream signals CHOP, eIF2α (and its phosphorylation), related molecular chaperones HSP10, 60, and 75, and matrix protease LONP1 at D1, D3, and D5 in C2C12 myoblasts treated with UCF101 (20 μM) or untreated, and quantification of WB assays of CHOP and p-eIF2α relative to GM (*n* = 3). LONP1, lon peptidase 1; CHOP (Ddit3), DNA-damage-inducible transcript 3; HSP, heat shock protein; eIF2α, eukaryotic translation initiation factor 2 subunit alpha; UPRmt, mitochondrial unfold protein response; Clpx, caseinolytic mitochondrial matrix peptidase chaperone subunit X; Clpp, aseinolytic mitochondrial matrix peptidase proteolytic subunit. Student’s *t*-tests were used. Statistical significance: * *p* ≤ 0.05, ** *p* ≤ 0.01, *** *p* ≤ 0.001.

**Figure 8 ijms-23-11761-f008:**
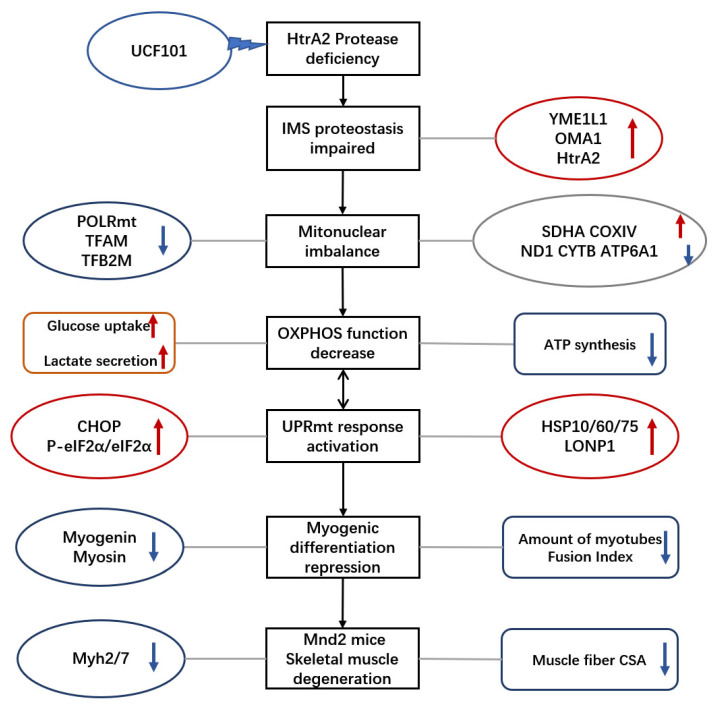
Mode picture. The arrows and circular marked in red: these indicators were upregulated The arrows and circular marked in blue: these indicators were downregulated.

**Table 1 ijms-23-11761-t001:** Primer Sequences for genotype identification.

Primers 5′-3′		
Gene	Forward	Reverse
*Mnd2*	CACACTGAGGATTCAAACCAAGGT	GACCGAGGACATAAACAGGGTGTA

**Table 2 ijms-23-11761-t002:** Primer Sequences for mRNA expression assay.

Primers 5′-3′		
Gene	Forward	Reverse
*GAPDH*	GTCGTGGAGTCTACTGGTGTC	GAGCCCTTCCACAATGCCAAA
*MyoD*	GACAGGGAGGAGGGGTAGAG	TGCTGTCTCAAAGGAGCAGA
*myogenin*	CCTACAGACGCCCACAATCT	CAGGGCTGTTTTCTGGACAT
*Myh1*	CTCTTCCCGCTTTGGTAAGTT	CAGGAGCATTTCGATTAGATCCG
*Myh2*	GCACCCATCCTCATTTCGTGA	GGAATGGCACTTGCGTTTAACA
*Myh4*	CTTTGCTTACGTCAGTCAAGGT	AGCGCCTGTGAGCTTGTAAA
*Myh7*	ACTGTCAACACTAAGAGGGTCA	TTGGATGATTTGATCTTCCAGGG
*Lonp1*	CTCATGGTGGAGGTTGAGAATG	CAGAGGGTTCAAGGCGATGAT
*Clpp*	CCATCTACGACACAATGCAGT	GAATTGGGCAGTGAATGGCG
*Clpx*	CCAGGCTGGATATGTAGGTGA	GCACACTGCCAATCTTATCTACT
*Chop*	AAGCCTGGTATGAGGATCTGC	TTCCTGGGGATGAGATATAGGTG
*Hsp10*	TTCCGCTCTTTGACAGAGTATTG	TCTGGAAGCATAATGCCACCT
*Hsp60*	GCCTTAATGCTTCAAGGTGTAGA	CCCCATCTTTTGTTACTTTGGGA
*HSP75*	CAGGACAGTTATACAGCACACAG	TACAGAGAACGGGCTACGATG

**Table 3 ijms-23-11761-t003:** Primer Sequences for mitocopy number assay.

Primers 5′-3′		
Gene	Forward	Reverse
*β-globin*	GAAGCGATTCTAGGGAGCAG	GGAGCAGCGATTCTGAGTAGA
*mNADH1*	TCTGCCAGCCTGACCCATAG	CCGGCTGCGTATTCTACGTT
*mCOXII*	GCCGACTAAATCAAGCAACA	CAATGGGCATAAAGCTATGG
*mCYTB*	TATTCCTTCATGTCGGACGA	AAATGCTGTGGCTATGACTG

## Data Availability

Not applicable.

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
