# Peer review of "Inhibition of High-Temperature Requirement Protein A2 Protease Activity Represses Myogenic Differentiation via UPRmt"

_ijms, 2022, doi:10.3390/ijms231911761_

Round 1

Reviewer 1 Report

The authors of the current manuscript investigated the role of mitochondrial intermembrane space (IMS) proteostasis during myogenic differentiation, through the evaluation of inhibitory effects of UCF101, an inhibitor of High-temperature-requirement protein A2 (HtrA2) during the differentiation process. The manuscript us generally well designed and reported. I have only a few comments below:

-        - In the last paragraph of introduction (line 90): It is not preferred to mention your results at the introduction section. Just state your aims and methodology synopsis.

-         - The institutional review board statement is missing at page 18.

-          -The inhibitor used UCF101 has been previously used in many manuscripts as a specific inhibitor of the HtrA2 protease; however, adding some of the most distinguished characteristics of the compound to the methods section, such its IC50 in comparison to the used concentration or its specificity criteria with other serine proteases would be useful.

-       -   Methodology (line 404): The number of animals in each group should be mentioned in methodology and the number of animals used in each experiment should be commented upon in the legends of the relevant figures.

-       - Figures 1,3,5,6, and 7: All WB images were provided without quantitation diagrams. Diagrams were only provided for RT-qPCR assessments. Visual inspection of WB is extremely important; however, the quantitation of band density in reference to the house keeping gene is also important to evaluate the significance of the findings and reproducibility in repetitive experiments. Quantitation can be performed easily with the ImageJ software. And the number of experiments of each WB analysis should be mentioned in the figure legend, since only a representative image for each experiment is put in the figure.

Author Response

Dear Reviewer:

Thanks for your letter and for the reviewers’ comments concerning our manuscript entitled “Inhibition of high-temperature requirement protein A2 protease activity represses myogenic differentiation via UPRmt”. (ID: ijms-1874754). Those comments are all valuable and very helpful for revising and improving our paper, as well as the important guiding significance to our researches. We have studied comments carefully and have made correction which we hope meet with approval. Revised portion are marked in the manuscript. We have shorten the Introduction part, quantified all key proteins and adjusted the composition of figures. The main corrections in the paper and the responds to the reviewer’s comments are as flowing:

Responds to the reviewer’s comments:

Major comments:

  1. - In the last paragraph of introduction (line 90): It is not preferred to mention your results at the introduction section. Just state your aims and methodology synopsis.

Response:

We are very grateful for the reviewer’s opinion. According to your suggestion, we have rewritten the last paragraph of the introduction.

  1. - The institutional review board statement is missing at page 18.

Response:

We are very sorry for the negligence in our manuscript and we have added the institutional review number and laboratory animal use license number in the prescribed place.

  1. -The inhibitor used UCF101 has been previously used in many manuscripts as a specific inhibitor of the HtrA2 protease; however, adding some of the most distinguished characteristics of the compound to the methods section, such its IC50 in comparison to the used concentration or its specificity criteria with other serine proteases would be useful.

Response:

Thanks for the reviewer’s suggestion. We have added the characteristics of the compound in the manuscript in both result and method section, UCF101 has been reported to have a significant inhibitory effect on HtrA2 protease activity at a concentration of 20 μM in previous studies [1-3]. This research does not involve the anti-apoptotic function of UCF101. In addition, we consulted the relevant literature before starting the experiment, so we selected drug concentrations that do not affect cell activity and apoptosis.

[1] Cilenti L, Lee Y, Hess S, et al. Characterization of a novel and specific inhibitor for the pro-apoptotic protease Omi/HtrA2. J Biol Chem. 2003;278(13):11489-11494.

[2] Regulation of the HTRA2 Protease Activity by an Inhibitory Antibody-Derived Peptide Ligand and the Influence on HTRA2-Specific Protein Interaction Networks in Retinal Tissues. Biomedicines. 2021;9(8):1013.

[3] Klupsch K, Downward J. The protease inhibitor Ucf-101 induces cellular responses independently of its known target, HtrA2/Omi. Cell Death Differ. 2006;13(12):2157-2159.

  1. -Methodology (line 404): The number of animals in each group should be mentioned in methodology and the number of animals used in each experiment should be commented upon in the legends of the relevant figures.

Response:

We are very sorry for the negligence in our paper and we have added the number of animals in each genotype in the prescribed place.

  1. - Figures 1,3,5,6, and 7: All WB images were provided without quantitation diagrams. Diagrams were only provided for RT-qPCR assessments. Visual inspection of WB is extremely important; however, the quantitation of band density in reference to the house keeping gene is also important to evaluate the significance of the findings and reproducibility in repetitive experiments. Quantitation can be performed easily with the ImageJ software. And the number of experiments of each WB analysis should be mentioned in the figure legend, since only a representative image for each experiment is put in the figure.

Response:

Thanks for the reviewer’s kindness. We added the quantitation analysis of the WB result expect the WB of Myosin in Figure 3a. Because there is almost no expression in GM group, the expression of Myosin is clearly in D3 and D5 both in CON and UCF101 group. Now, all key protein mentioned in our paper has been quantified, and the number of experiments of each WB analysis has been mentioned in the figure legend.

In addition, we have consulted with a professional English editor and revised the manuscript. We would like to thank the reviewer again for taking the time to review our manuscript.

Reviewer 2 Report

Hongyu Sun et co-Authors, in the manuscript entitled “Inhibition of high-temperature requirement protein A2 protease activity represses myogenic differentiation via UPRmt”, aim to investigate the role of HtrA2 in the IMS proteostasis during myogenic differentiation. They used two different model systems: the already reported mnd2 mice and C2C12 mice myoblasts.  UCF101, a specific inhibitor of HtrA2, was evaluated to assess the role of the protease during the C2C12 cells' myogenic differentiation. They found that during the imbalance of IMS proteostasis, myogenic differentiation was significantly repressed, oxidative phosphorylation was defective, followed by an ETC complex impairment. Further, CHOP, p-eIF2α, and other mitochondrial unfolded protein response (UPRmt)-related proteins were upregulated. They conclude that the imbalance of mitochondrial IMS proteostasis acts as a retrograde signaling pathway to inhibit myogenic differentiation through UPRmt.

Overall, I find the data of some interest, and the manuscript might be relevant for the field. I have the following comments:

-in the Introduction section it should be stated that two different models are used in the work.

-the Authors say that: “a widely accepted hypothesis is that the depletion of HtrA2 activity impairs mitochondrial function and activates UPRmt by disrupting mitochondrial proteostasis and increasing ROS production [24,50-52]. However, we did not find upregulation of UPRmt-related molecules in mnd2 mice in our earlier study [17], which we believe may be related to the different physiological state of myoblasts during differentiation as compared to mature skeletal muscle cells.” This point is crucial for the work and an alternative hypothesis should be suggested.

-As far as the appropriateness of figures, I have the following main concerns:

-In fig.1a, it is not clear the identity of each PCR product.

-Fig.1ef, why there are two bands for myogenin, and what is the MW of the other bands that are not indicated.

-In fig. 1 g, the pattern of TFB2M is not clear and visible, especially in the homozygous mouse

-Fig.1h, it is not indicated the reason there are several bands for Oma1. The HtrA2 signals are both stronger in HtrA2-/- compared to wt samples but this is not for Yme1L1

-In fig. 5 the signals of cytb, SDHa, PGC1a, and NRF2 are dual, and it is not indicated which form has been considered in the experiments and how have been estimated.

-Overall, in the Figures and in the Results section, it is not indicated which form has been considered in the experiments and how they have been estimated.

-It should be further strengthened and discuss the correlation between the decrease of mtDNA copy number and of the mtDNA-encoded subunits (ND1, cytb, ATP6)

-In the Introduction, it is said that the “key molecules related to oxidative phosphorylation function, the ETC complexes……”. Please, correct this because the ETC complexes are not molecules.

In all the wb experiments where duplicates are present, they are not specified.

In general, I find that the duplication of a sample in the same gel is not sufficient to say that the result is replicable. Each experiment should be replicated at least three times performing three independent gel runs and with samples coming from different experiments. All the bands considered to establish the effect of the drug should be quantified.

Author Response

Dear  Reviewer:

  Thanks for your letter and for the reviewers’ comments concerning our manuscript entitled “Inhibition of high-temperature requirement protein A2 protease activity represses myogenic differentiation via UPRmt”. (ID: ijms-1874754). Those comments are all valuable and very helpful for revising and improving our paper, as well as the important guiding significance to our researches. We have studied comments carefully and have made correction which we hope meet with approval. Revised portion are marked in the manuscript. We have shorten the Introduction part, quantified all key proteins and adjusted the composition of figures. The main corrections in the paper and the responds to the reviewer’s comments are as in the PDF file.

Round 2

Reviewer 2 Report

The manuscript has been greatly improved at several levels and the Authors answered the main point of criticism.